# Comparative Review of the Current and Future Strategies to Evaluate Bone Marrow Infiltration at Diffuse Large B-Cell Lymphoma Diagnosis

**DOI:** 10.3390/diagnostics14060658

**Published:** 2024-03-21

**Authors:** Fernando Martin-Moro, Javier Lopez-Jimenez, Jose A. Garcia-Marco, Jose A. Garcia-Vela

**Affiliations:** 1Hematology Department, Ramon y Cajal University Hospital, 28034 Madrid, Spain; 2Doctoral Program in Health Sciences, University of Alcala, 28801 Alcala de Henares, Spain; 3Hematology Department, Puerta de Hierro University Hospital, 28222 Majadahonda, Spain

**Keywords:** bone marrow, large B-cell lymphoma, lymphoma, histology, flow cytometry, PET

## Abstract

Diffuse large B-cell lymphoma (DLBCL) requires a complete staging at diagnosis that may have prognostic and therapeutic implications. The role of bone marrow (BM) biopsy (BMB) is controversial in the era of nuclear imaging techniques. We performed a comparative review of 25 studies focused on BM evaluation at DLBCL diagnosis, including at least two of the following techniques: BMB, flow cytometry, and positron emission tomography (PET-FDG). The report about BM involvement (BMi), diagnostic accuracy, and prognostic significance was collected and compared among techniques. A concordance analysis between BMB, FCM, and PET was also performed, and we deeply evaluated the implications of the different types of BMi: concordant by LBCL or discordant by low-grade B-cell lymphoma for both BMB and FCM, and focal or diffuse uptake pattern for PET. As a main conclusion, BMB, FCM, and PET are complementary tools that provide different and clinically relevant information in the assessment of BMi in newly diagnosed DLBCL.

## 1. Introduction

Diffuse large B-cell lymphoma (DLBCL) is the most common lymphoma [1]. It is a biologically heterogeneous disease with aggressive behavior that requires prompt chemoimmunotherapy administration after diagnosis. Prior to treatment initiation, a complete staging assessment must be performed to evaluate the disease extension [2], which is also an important aspect necessary to calculate the patient risk and may be a key for designing the therapeutic approach. Thus, in most patients with localized stage, a frontline approach with a reduced number of chemoimmunotherapy cycles with or without radiotherapy is enough to achieve a complete remission [3,4], and cases with advanced stage and intermediate to high-risk patients probably benefit from receiving Pola-R-CHP (polatuzumab vedotin, rituximab, cyclophosphamide, doxorubicin, and prednisone) as first-line therapy rather than R-CHOP (rituximab, cyclophosphamide, doxorubicin, vincristine, and prednisone) [5].

When it comes to addressing DLBCL extension, whole-body positron emission tomography with 18-fluorodeoxiglucose along with computerized tomography (PET-FDG/CT) is the key technique to perform the staging evaluation [2]. The examination of the central nervous system by imaging (magnetic resonance) or cerebrospinal fluid analysis (flow cytometry and cytomorphological assessment) is recommended in cases considered high-risk of neurologic invasion [6], such as those with high CNS-IPI or patients with extranodal involvement of certain sites such as the testes or kidneys. In reference to bone marrow (BM) evaluation, according to the last Lugano criteria [2] a morphologic and immunohistochemical examination of the BM trephine biopsy (BMB) in patients with DLBCL is only needed when BM infiltration (BMi) is negative by positron emission tomography (PET), and the identification of occult discordant histology is clinically important.

BMi assessment can be performed using different techniques. BMB has been historically considered the gold standard for detecting BMi in DLBCL [7] until the development of nuclear imaging assays, but nowadays there is a hot debate about the diagnostic accuracy and prognostic role of both techniques. BMB is an invasive procedure that may bring complications such as pain, anxiety, or bleeding [8,9]. In some cases, the BM sample obtained from the iliac crest is not sufficient for morphological assessment, and in others, the BMB analysis yields a false negative result due to focal marrow infiltration in a location different from the punctured one [10]. The characterization of BMi by BMB in DLBCL includes two types of morphological invasion: concordant (BMi by LBCL) and discordant (BMi by small cell low-grade B-cell histology). PET with computed tomography (PET-CT) is a non-invasive metabolic and imaging technique that has demonstrated a high sensitivity in detecting BMi in the setting of aggressive lymphomas [11]. Nevertheless, some authors find it controversial to avoid BMB in most DLBCL cases, as the 2014 Lugano criteria suggest [12,13,14]. When it comes to evaluating BM characterization by PET, different uptake patterns have been studied, with the focal and diffuse ones being the most widely described [15]. A focal uptake is usually defined as one or more circumscribed areas of high fluorodeoxyglucose (FDG) uptake within the skeleton or marrow space, and diffuse uptake is considered the uniformly increased FDG uptake throughout the marrow space. In patients with DLBCL, a focal FDG uptake in the BM has been described as a pattern of tumor infiltration in most cases; however, the significance of a diffuse uptake is controversial, being related to reactive, inflammatory, or non-malignant conditions in some cases [16,17,18]. Furthermore, it has been shown that PET is less sensitive in detecting BMi by low-grade B-cell lymphoma and concordant LBCL BMi of low quantity [19,20]. The role of some other technologies with higher methodological sensitivity rates has been evaluated in the setting of assessing BMi in DLBCL, such as flow cytometry (FCM) or molecular assays. Prior recommendations in response criteria for malignant lymphoma proposed that, for routine practice clinical decision-making, only BMi greater than 2% by FCM or genetics should be taken into account if BMB is negative [21]. FCM assessment has been suggested to be complementary to BMB in detecting BMi at DLBCL diagnosis [22], although it is not exempt from technical difficulties for detecting concordant involvement, such as hemodilution of the BM aspirate or cellular adhesion in bone trabeculae [23,24]. Nonetheless, FCM has demonstrated an important role in detecting discordant low-grade BMi in DLBCL [25,26].

When it comes to defining the prognostic significance of BMi at DLBCL diagnosis, both the technique and the infiltration pattern seem to be relevant. BMB has demonstrated being an adverse prognostic factor, even independently from the International Prognostic Index (IPI), performance status, or age [27,28,29]. The concordant infiltration of the BM has clearly shown a prediction of a worse prognosis when compared to the discordant BMi [30,31], but there is no agreement if the discordant pattern implies an intermediate prognosis between the concordant one and the absence of BMi [27,28,32] or if discordant cases have a similar outcome to patients without BMi [29,31,33]. The role of PET in defining prognosis in DLBCL, according to BMi, is controversial. Some studies suggest that BMi by PET is associated with worse outcomes [16,34], while other studies indicate that it has no prognostic relevance [35,36]. In the setting of BM evaluation by PET, it is described that the focal uptake pattern implies a worse prognosis than the diffuse pattern [16], whose outcome could be equivalent to patients with negative BMi by PET. The prognostic impact of FCM or molecular assays has scarcely been reported.

On many occasions, discrepancies between the different techniques occur when assessing BMi in DLBCL, and the diagnostic accuracy and definition of a true positive BMi are still controversial. Furthermore, the surveillance implication of BMi is still not clear, and some studies have combined and compared the results of different techniques in an attempt to solve this topic. Our aim is to perform a comparative review of the three techniques most widely performed when evaluating BMi at DLBCL diagnosis: BMB, PET, and FCM. We analyzed the rate of BMi by each technique and their diagnostic accuracy, described the concordance and discordance among procedures in detecting BMi, and evaluated the prognostic implication of the BMi by each technique when compared or combined with the others. When appropriate and available, data about each type of result were also collected, thus concordant or discordant BMi by BMB and FCM and focal or diffuse pattern of BMi by PET-FDG.

## 2. Materials and Methods

### 2.1. Population

The study population included patients with a histological diagnosis of DLBCL with a BM assessment at diagnosis, including a direct BM evaluation (BMB and/or FCM) with or without a nuclear-imaging assessment (PET-FDG), and prior to treatment initiation. Only adult patients (aged 18 or greater) and those treated in the rituximab era (mainly with R-CHOP) were considered.

### 2.2. Bibliographic Search Strategy and Selection

MEDLINE and Embase were used for the study search. The search strategy was *(marrow[Title] OR BM[Title]) AND (DLBCL[Title] OR LBCL[Title] OR diffuse large[Title] OR large B-cell[Title] OR large B cell[Title])* for MEDLINE and *‘marrow’:ti AND (‘dlbcl’:ti OR ‘large b cell:ti’ OR ‘large b-cell’:ti OR ‘lbcl’:ti) AND [embase]/lim AND [humans]/lim AND*
*[2000–2024]*/*py* for Embase. Other sources, such as article citations, were also used. The study period was from January 2000 to December 2023.

After eliminating duplicates, two authors (F.M-M. and JA.G-V.) screened the records by title and abstract, removing reports due to article type (reviews, case reports, case series, commentaries, responses, and conference abstracts) or because studies did not involve human samples, full-text was not available, histologies included were different than DLBCL, or research was not focused on BMi assessment or prognosis. Then, the same authors evaluated full-text articles for their eligibility, eliminating those because of research not performed at diagnosis, a small sample (less than 75 patients), pre-rituximab era, imaging or genetics-based studies, or insufficient reporting. One important point is that only articles were considered suitable for performing comparisons—and therefore to be included in this review—if at least results from two of the three chosen techniques (BMB, FCM, and PET) were reported. Articles based on magnetic resonance imaging (MRI) or genetic assays were excluded due to the scarce presence of comparative studies with other techniques in the literature (less than five of each of them).

### 2.3. Statistical Analysis

A flow chart was created to illustrate the process of bibliographic search and article selection. A description of each included study and their population characteristics was performed, including region, study type, recruitment period, sample size (N), male/female ratio, age, cell-of-origin (COO) classification according to the Hans algorithm, stage, IPI, frontline approach, and follow-up time. The BM assessment according to BMB, FCM, and/or PET was included according to the reports of each report, describing separately the type of BMi (concordant or discordant for BMB and FCM and focal or diffuse patterns for PET) if available. A *complete* report was considered if information about both concordant and discordant BMi by BMB or FCM was available and if information about both focal and diffuse BM uptake patterns by PET was available. Proportions of *complete* and *incomplete* reports among each technique were compared by the chi-square test or Fisher’s test when appropriate. A complete record of the number of patients with BMi by each technique and type of infiltration was calculated to assess the proportion of involvement among the global cohort, making comparisons between techniques and infiltration patterns by chi-square test. Statistical significance was considered when *p* values were lower than 0.05.

An analysis of concordance and discordance between the different reported techniques was performed. Cohen’s kappa index was obtained from records or calculated, if not reported, to measure the statistical concordance among techniques to evaluate BMi at DLBCL diagnosis. Results were interpreted as follows: values ≤ 0 indicated no agreement, 0.01 to 0.2 as none to slight agreement, 0.21 to 0.4 as fair, 0.41 to 0.6 as moderate, 0.61 to 0.8 as substantial, and 0.81 to 1 as almost perfect agreement.

If information about sensitivity, specificity, positive predictive value, negative predictive value, Youden’s index, or diagnostic accuracy was reported by any of the included articles, it was also recorded. As the definition of a true positive BMi may change among studies, it was also described for each report.

Direct or indirect prognostic comparisons were described according to the information available from studies reporting the outcome significance of BMi at DLBCL diagnosis using at least two techniques. If the prognostic evaluation was performed on a subcohort of patients in each study, it was specified. Survival endpoints included progression-free survival (PFS, time from diagnosis to first disease progression or relapse) or event-free survival (EFS, time from diagnosis to first disease progression or relapse or death by any cause), as reported by each study, and overall survival (OS, time from diagnosis to death by any cause). It was described as the available information about prognosis and univariate/multivariate hazard ratio analysis (Cox regression model) of each technique and/or type of BMi. The other variables studied in each reported multivariate analysis, which included BM assessment, were also recorded.

## 3. Results

### 3.1. Description of the Included Studies, Characteristics of the DLBCL Populations, and BM Evaluation

A total of 369 unique citations were identified from the electronic database and other sources searched. Of these, 291 records were excluded based on title and abstract screening, and 53 were excluded after full-text evaluation. In the end, 25 studies were included in the comparative review (Figure 1), accounting for 4849 newly diagnosed DLBCL patients. Study characteristics are presented in Table 1. Almost all records were retrospective studies (22/25, 88%), 2/25 were prospective (8%), and 1/25 did not report the type of study (4%). All studies provided information about BMB, while 6/25 (24%) included FCM (accounting for 814 patients), and 21/25 (84%) provided PET data (accounting for 4175 patients). Only 2/25 (8%) combined BMB, FCM, and PET; 4/25 (16%) included BMB and FCM; and 19/25 (76%) reported BMB and PET. No article combining solely FCM and PET was found. PET reports were *complete* of BM characterization at a higher rate than BMB (71% vs. 40%, *p* = 0.033) and FCM (71% vs. 17%, *p* = 0.027) ones (Figure 2). Although proportional differences were seen in the rate of *complete* reports between BMB and FCM, statistical significance was not reached (40% vs. 17%, *p* = 0.38).

BM direct assessment was performed in 20/25 studies (80%) after unilateral iliac crest punction; in 4/25 (16%), it was performed bilaterally; and 1/25 (4%) reported both results after unilateral and bilateral procedures; this study [37] accounted for the rate of BMi twice, by BMB and FCM, separating the results from unilateral and bilateral iliac crest punctions. In the global cohort, BMB detected BMi in 15% of DLBCL patients (744/4951), with a slightly higher rate of BMi detected by bilateral punction (17.1%, 244/1425) than by unilateral punction (14.8%, 480/3255) (*p* = 0.039). A concordant and discordant BMi by BMB were seen in 11.5% (238/2062) and 4.7% (88/1978) of patients, respectively. FCM showed BMi in 24.5% of cases, a higher rate when compared with BMB (*p* < 0.001). Only one study [47] provided sufficient information about the type of BMi by FCM, detecting concordant involvement in 10.3% of cases (24/232) and discordant in 15.1% (35/232). There was no difference between BMB and FCM in detecting a concordant invasion of the BM (*p* = 0.59), but it was found when assessing a discordant infiltration favoring the FCM analysis (*p* < 0.001). BM uptake by PET-FDG was globally described in 21.9% of DLBCL cases (749/4175 included in 18 studies); a focal and diffuse pattern of BM uptake was seen in 18.1% (714/3949 included in 18 studies) and 5.5% (174/3191 included in 15 studies) of patients, respectively.

*Summary of the section*: All reports, including at least two techniques for the evaluation of BMi at DLBCL diagnosis, reported information about BMB. FCM and PET showed higher rates of BMi (24.5% and 22%, respectively) than BMB (15%).

### 3.2. Concordance Analysis among BMB, FCM, and PET When Assessing BMi at DLBCL Diagnosis

All 25 studies were suitable for concordance comparisons between the assessed techniques (Table 2).

#### 3.2.1. Concordance between BMB and FCM in the Setting of DLBCL BMi Assessment

Five studies evaluated the concordance between BMB and FCM. The median Cohen’s kappa index of the studies was 0.65 (range 0.25–0.66), which is defined as a substantial agreement between techniques. The proportion of concordant positive (+) and negative (−) cases for both BMB/FCM ranged 3–26% and 37–83%, respectively. The most common discordant rates were seen favoring BMi by FCM; thus, the BMB−/FCM+ group accounted for 7–37% of cases, while only 0–6% of cases were classified as BMB+/FCM-. Interestingly, one study [49] reported that among FCM+ patients, the percentage of pathological cells infiltrating the BM detected by FCM was higher among BMB+ cases compared with the BMB− ones (15% vs. 2.1%, *p* < 0.001).

#### 3.2.2. Concordance between BMB and PET in the Setting of DLBCL BMi Assessment

Twenty-one studies evaluated the concordance between BMB and PET. The median Cohen’s kappa index of the studies was 0.36 (range 0.19–0.68), which is defined as a fair agreement between techniques. Concordant BMB+/PET+ and BMB−/PET− cases accounted for 5–21% and 50–83%, respectively. Discordant results were seen in both ways, thus in up to 17% of cases, BMi is detected only by BMB (BMB+/PET−) and in up to 29% only by PET (BMB−/PET+). In one study [37], the agreement among techniques changed from moderate (Cohen’s kappa index 0.59) to substantial (Cohen’s kappa index 0.8) after performing a second BMB in cases of BMB−/PET+. Another study [10] showed that the agreement rose from none-slight (Cohen’s kappa index 0.19) to fair (Cohen’s kappa index 0.23) when considering only PET+ cases with focal uptake patterns. As described in different studies, a cause of BMB−/PET+ discrepancy is due to focal involvement far from the iliac crest location [10]; in fact, the focal pattern is described as the most common in BMB−/PET+ cases [35,36]. In contrast, there is no agreement among articles in defining the most common uptake pattern in BMB+/PET+ cases, being mainly focal in some studies [20], while diffuse in others [36]. When it comes to defining the type of infiltration by BMB, different studies agree that concordant involvement is mostly found in BMB+/PET+ cases, while patients from the BMB+/PET− group usually have a discordant invasion or a low-quantity concordant one [20,33,36]. One study [38] suggests higher rates of BMB/PET discrepancies among elderly patients and those with high-grade B-cell lymphoma with MYC and BCL2 and/or BCL6 rearrangements.

In one study [35], if a BMB+/PET− case was found, investigators reviewed PET images to check for BMi. In contrast, in some studies, a guided biopsy of the location with BM uptake was performed in BMB−/PET+ patients, proving BMi by BMB in most cases [33,37,41].

#### 3.2.3. Concordance between FCM and PET in the Setting of DLBCL BMi Assessment

Only two studies evaluated the concordance between FCM and PET. As described in the BMB/PET analysis, in one study [37], the agreement among FCM and PET rose from moderate (Cohen’s kappa index 0.54) to substantial (Cohen’s kappa index 0.74) after performing a second BM punction in patients with BMB−/PET+. The other study [43] reported a fair agreement between techniques (Cohen’s kappa index 0.3). Both articles agree in describing that approximately 70% of cases are both FCM/PET negative, while 8–14% are FCM+/PET+. Discrepant cases accounted for 4–13% for FCM+/PET− and 8–12% for FCM−/PET+.

*Summary of the section*: Substantial concordance has been described between BMB and FCM in evaluating BMi at DLBCL diagnosis, while concordance between BMB and PET was fair. Discordance among techniques was common.

### 3.3. Diagnostic Accuracy of BMB and PET for Detecting BMi at DLBCL Diagnosis

Few studies reported sufficient information about sensitivity, specificity, positive predictive value, negative predictive value, Youden’s index, or diagnostic accuracy (Table 3). Furthermore, the definition of a true positive BMi strongly varies among studies. It seems that PET sensitivity is higher when BM is focal rather than diffuse [10], which indicates that some studies considered true positive cases for BMi those with PET+ by a focal pattern [16,33,44]. Studies that used a true positive BMi defined by BMB+ or by PET+ if accompanied with targeted/morphologic imaging changes or FDG disappearance with response after treatment or concordant FDG progression on follow-up [10,44,46], reported a higher PET sensitivity and diagnostic accuracy, while a higher BMB specificity. Equivalent findings were seen in one study that considered BMB+ or PET+ focal as true positive BMi [16]. As expected, when BMB+ was considered solely as true positive BMi, the sensitivity, specificity, predictive value, and diagnostic accuracy of PET were reduced [33,37,41,42,45,51], as it also happened to BMB when true positive BMi was only defined by PET+ [33]. No studies reported data about sensitivity or specificity regarding BMi by FCM.

The upstaging to a higher stage by BMB or PET has been evaluated in many studies. Discrepant results were seen about the role of BMB in this setting; thus, some studies described that BMB upstaged to Ann Arbor IV in 2–15% of cases in their global cohorts that were grouped as PET− for BMi [33,42,50], while others reported that BMB did not upstage any BMB+/PET− case to Ann Arbor IV [20,35,44]. Four studies agreed in reporting that up to 5–8% of patients in their global series were upstaged to Ann Arbor IV due to BMi by PET (particularly the focal pattern) not detected by BMB [10,20,35,50].

*Summary of the section*: PET-FDG showed higher sensitivity and BMB higher specificity when assessing BMi at DLBCL diagnosis, although high heterogeneity was seen among reports.

### 3.4. Prognostic Impact of BMi at DLBCL Diagnosis According to BMB, FCM, and PET in Comparative Studies

Eighteen studies, accounting for 3649 patients, evaluated the prognostic impact of BMB, FCM, or PET by comparing or combining at least two techniques (Table 4). There were no studies comparing the PET and FCM findings.

#### 3.4.1. Outcomes According to BMi by BMB and FCM in Newly Diagnosed DLBCL

Four studies (536 patients) described outcomes according to BMB and FCM. Despite the low number of reports, it seems that BMi by FCM is associated with a worse prognosis when compared with FCM− cases, even in patients with BMB+. Two studies [43,49] that analyzed outcomes after combining BMB/FCM findings described discrepancies about the worse prognosis of BMB+/FCM+ cases compared with BMB−/FCM+ ones. On the one hand, M. Moro et al. reported that the BMB−/FCM+ group presented an adverse outcome equivalent to the BMB+/FCM+ population; similar findings were described in another study not included in the comparative review due to insufficient data [56]. On the other hand, Greenbaum et al. described that the prognosis of BMB−/FCM+ cases was intermediate between the BMB−/FCM− and the BMB+/FCM+ groups. As a third study points out [40], both BMB and FCM analyzed separately are related to a worse prognosis, with a trend toward a worse outcome when BMi is detected by BMB.

Two studies that did not meet the criteria to be included in the comparative review [32,57] and one more recent study that was included [47] analyzed the prognostic role of BMi detected by direct BM assessment when combined with the cell-of-origin classification according to the Hans algorithm. The first study reported that BMB+ cases presented an equivalent worse prognosis regardless of being classified as germinal center B (GCB) or non-GCG [32], while in the second study, non-GCB BMB+ cases presented a worse outcome than the GCB BMB+ group [57]. Interestingly, the third study suggested that BMB+/FCM+ with a concordant BMi presented a worse outcome regardless of cell-of-origin classification, in contrast with BMB+/FCM+ cases with a discordant type of BMi, in which non-GCB patients presented a worse prognosis when compared with GCB cases [47].

#### 3.4.2. Outcomes According to BMi by BMB and PET in Newly Diagnosed DLBCL

Fourteen studies (3113 patients) reported surveillance information according to BMB and PET.

Most reports (9/14) described a worse prognosis when BMi was detected by BMB than by PET. Seven studies [10,35,36,45,50,51,53] reported worse outcomes (PFS/EFS and/or OS) in BMB+ DLBCL patients than in PET+ ones, and similar results were seen in another article analyzing PET+ focal cases [46]. Another group [48] ranked patients according to their BMi by BMB/PET from worse to better prognosis as follows: BMB+/PET+, BMB+/PET−, and BMB−/PET+; the BMB−/PET+ group presented an equivalent prognosis to BMB−/PET− cases. In contrast, Hong et al. reported similar outcomes between the BMB+/PET+ and the BMB+/PET− groups, while the surveillance was still equivalent between BMB−/PET+ and BMB−/PET−. Four studies out of 14 showed no differences between BMB and PET in predicting prognosis by BMi [20,41,48,52]: Liam et al. and Liang et al. reported equivalent prognosis between BMB+ and PET+, and Wang et al. described similar findings between BMB+ and PET+ focal; Cerci et al. saw that BMB+/PET+ focal patients associated a worse outcome when compared with BMB+/PET−, BMB−/PET+, and BMB−/PET+ focal, without differences between these three groups. Only 1/14 reports showed a worse outcome in PET+ focal cases than in BMB+, although it was only described for PFS and not for OS [16]. Additionally, one analysis including only stage IV DLBCL patients showed no differences in EFS or OS between the BMB+ or PET+ group and the PET+ group [33].

Regarding the BM uptake pattern by PET, three studies reported the prognostic significance of PET+ focal [20,46,48]. Chen et al. presented data about prognosis according to focal and diffuse PET+ patterns, showing PET+ focal cases had a worse impact in both EFS and OS, while PET+ diffuse cases did not relate to a worse outcome. Only one study described a slight trend toward worse lymphoma-specific survival in PET+ diffuse patients than in PET+ focal ones [41].

No report studied the prognostic significance of BMB according to concordant or discordant types of infiltration in the setting of a comparative study with BMi by PET.

*Summary of the section*: BMi detected by FCM is associated with worse outcomes in newly diagnosed DLBCL, even independently from BMB findings. The prognostic impact of BMi by PET is less clear.

## 4. Discussion

This is the first comparative review evaluating the diagnostic accuracy and prognostic implications of three different techniques (BMB, FCM, and PET) regarding BMi at DLBCL diagnosis. This review included twenty-five studies with a total sample size of 4849 patients with newly diagnosed DLBCL. Although BMB data were available in all studies, complete information about BMi (defined as a description of the different types of BMi according to each technique) by PET was available in a higher proportion of reports. It is probably due to the growing interest in nuclear imaging techniques to evaluate BMi in an attempt to abandon the direct assessment of BM by trephine biopsy and aspiration in DLBCL patients. The global rate of BMi was 15% for BMB, 24.5% for FCM, and 21.9% for PET. This review also supports the fact that a unilateral punction of the iliac crest is sufficient for assessing BMi in DLBCL, whereas performing a bilateral BMB is not clinically relevant. The BMB detected concordant infiltration more frequently than the discordant one, while the sole study that reported information about the type of BMi by FCM observed a higher rate of discordant invasion [47], in line with previous reports that highlighted the role of FCM in detecting BMi in low-grade small B-cell populations in DLBCL [25,26]. In a concordance comparison between techniques regarding their role in detecting BMi at DLBCL diagnosis, the agreement between BMB and FCM was higher than between BMB and PET. Most discrepancies between BMB and FCM occur due to discordant or minor BMi detected by FCM and not detected by BMB, probably explained by the higher sensitivity of the FCM technique. In this setting, future studies based on FCM may provide interesting information about the biological implications of detecting discordant BMi by a low-grade B-cell entity in DLBCL patients; in fact, although the possible clonal relationship between the DLBCL and their BM discordant clones has barely been studied, it is a topic of high interest that may have diagnostic and therapeutic implications in the future. The high incidence of clonal identity in DLBCL cases with discordant BMi could suggest that in those cases, the DLBCL develops from a previous indolent B-cell lymphoma [47,58], and, in this setting, BM detection and complete phenotypic characterization of the low-grade population by FCM may provide a first step in the suspicion of a histological transformation. In fact, some patients with DLBCL are concomitantly diagnosed with indolent lymphoma, but their outcomes do not differ from those diagnosed with DLBCL alone [59].

When BMB and PET were compared, both BMB+/PET− and BMB−/PET+ discrepant cases were seen among the studies, suggesting that both techniques are complementary in assessing BMi in newly diagnosed DLBCL. BMB is probably a better tool for detecting concordant BMi of low quantity or discordant infiltration than PET, and PET-FDG is easily able to determine focal involvement that BMB cannot recognize without a guided punction. In fact, a focal pattern of BM uptake detected by PET is highly suggestive of BMi, and a guided BM assessment is not necessary to confirm the invasion of BM in these cases; however, even nowadays, there is no established definition of true positive BMi by PET in DLBCL. Some studies included as part of the definition of true BMi by PET the concept of following up the uptake patterns of the BM, making comparisons with the response or progression of the DLBCL in later explorations, and establishing the BMi if concordance between determinations is found, thus the disappearance of BM FDG uptake after treatment or BM FDG increasing with disease progression. Apart from the fact that a retrospective evaluation of BMi is not operative for providing this information at DLBCL diagnosis, evaluating the BM PET may be even more challenging during follow-up due to the high frequency of reactive uptake patterns, such as in the context of the systemic therapies administered (such as steroids, immunochemotherapy, or supportive treatment with granulocyte colony stimulating factor), driving false positive results. Nonetheless, as previously reported, PET-FDG is highly accurate for detecting BMi in newly diagnosed DLBCL, and BMB may be avoided in cases with positive PET BM uptake [15], but it is a fact that in some cases with negative PET, an occult BMi may be present, and a direct BM assessment may be performed to rule it out. Another controversial topic is the fact that avoiding BMB may or may not have staging implications and, consequently, a possible effect on the therapeutic approach. Since there are both studies that defend the role of BMB in upstaging to advanced disease DLBCL cases with negative PET for BMi [33,42,50] and others that report the null effect of BMB in upstaging PET negative cases [20,35,44], from the point of view of the authors, there is not enough strong evidence to affirm that abandoning BMB would not have clinically relevant consequences.

Studies describing DLBCL outcomes according to BMi when comparing different techniques mostly report data facing BMB and PET results. Although BMi by BMB has clearly demonstrated being an adverse prognostic factor in DLBCL, there is no sufficient evidence to determine if BM uptake by PET is associated with a worse prognosis in DLBCL patients with positive BMB. Even though some studies did not find a correlation between BMi by PET and a worse prognosis, this could be explained due to the different prognosis significance among the patterns of BM uptake (focal or diffuse). Two studies [41,50] described higher median SUVmax in DLBCL cases with BM focal patterns compared with those with BM diffuse patterns. Some groups are investigating the prognostic implications of metabolic measures in DLBCL patients, including parameters related to BM, such as the BM retention index and the BM-to-liver ratio of baseline PET-FDG, both described as predictors of PFS and OS [60]. Few studies have analyzed the prognostic impact of BMi by FCM in DLBCL, showing that a positive BMi detected by FCM is associated with a worse outcome, even without considering BMB findings. The fact that DLBCL patients with BMi demonstrated by FCM but not by BMB have a worse prognosis when compared with cases both negative for BMB and FCM is very relevant, as it may suggest that BMB is not able to screen a group of patients with a worse outcome in which a direct BM is performed. The combination of the type of BMi by BMB/FCM (concordant or discordant) and the immunohistochemical cell-of-origin classification is an interesting topic that should be evaluated in future studies, as it seems that non-GCB patients associate a worse prognosis if BMi is detected by direct BM assessment, while the prognosis of the GCB group depends on the type of BMi.

Some other tools have been explored to assess BMi at DLBCL diagnosis. The examination of pelvic [61] or whole-body [62] magnetic resonance imaging (MRI) focusing on DLBCL BMi has been evaluated as a complementary tool for BMB and PET, and even a prognostic implication of positive MRI has been suggested. Different genetic assays have been studied to detect BMi in de novo DLBCL. The molecular analysis of immunoglobulin heavy chain (IgH) gene rearrangement has been evaluated in some studies, combining their results with BMB [63] or PET-FDG [39] for better diagnostic accuracy and a greater prognostic stratification in newly diagnosed DLBCL cases. Furthermore, the finding of cytogenetic alterations by karyotyping or fluorescence in situ hybridization [64] and the detection of occult involvement by polymerase chain reaction [54] in BM assessment of DLBCL patients has been related to worse outcomes independently from BMB results.

## 5. Conclusions

BM assessment in newly diagnosed DLBCL is still a controversial and interesting topic, with discrepant results in the literature regarding the diagnostic accuracy and prognostic implications of the different available techniques for its evaluation. Both the BMB, the FCM, and the PET-FDG are complementary tools that provide different and clinically relevant information. There is still not enough evidence to recommend avoiding the direct BM assessment at the baseline evaluation of DLBCL. In fact, it is necessary to increase knowledge in this regard by performing prospective studies and deeply analyzing the role of novel technologies such as multiparametric FCM and molecular assays.

## Figures and Tables

**Figure 1 diagnostics-14-00658-f001:**
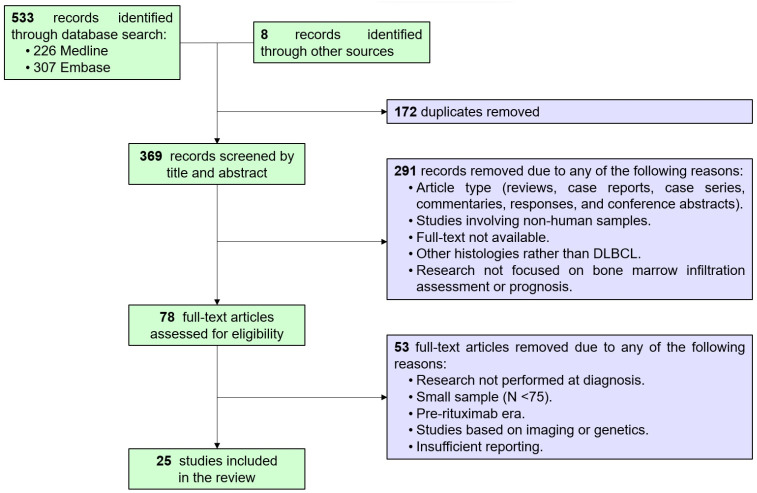
Flow chart presenting the study selection process.

**Figure 2 diagnostics-14-00658-f002:**
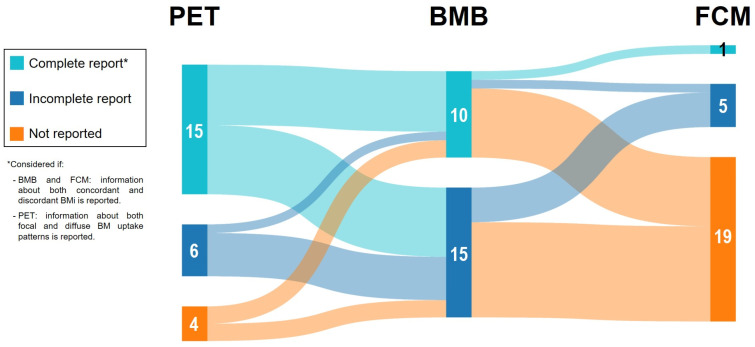
Correlation between techniques (PET/BMB and BMB/FCM) according to available data reported in the twenty-five studies included in the review.

**Table 1 diagnostics-14-00658-t001:** Characteristics and pretreatment bone marrow evaluation of newly diagnosed DLBCL patients.

Ref. *	Region	Study Type	Period	N	DLBCL Population Characteristics							Bone Marrow Assessment								
					Male/Female Ratio	Median Age (Range)	COO ^†^	Ann Arbor	IPI	Frontline	Follow-Up (Months)	Histology			Flow Cytometry			PET-FDG		
												Not Specified or Global	Concordant	Discordant	Not Specified or Global	Concordant	Discordant	Not Specified or Global	Focal	Diffuse
Bo et al. [37]	China	Retrospective	2019–2022	102	1.2	29.4% ≥ 60 yo	GCB: 35% Non-GCB: 65%	I–II: 14% III: 0% IV: 86%	3–5: 28%	-	-	21% (28% after second punction)	-	-	18% (24% after second punction)	-	-	26%	23%	3%
Han et al. [38]	South Korea	Retrospective	2014–2020	328	1.3	59 (44–74)	-	I–II: 52% III–IV: 48%	3–5: 41%	-	-	14%	-	-	-	-	-	18%	11%	7%
Kim et al. [39]	South Korea	Prospective	2017–2018	94	1.8	66 (24–85)	-	I–II: 47% III–IV: 53%	3–5: 61%	R-CHOP	35 (23–47)	10%	6%	3%	-	-	-	17%	12%	5%
Okamoto et al. [40]	Japan	Retrospective	2012–2018	221	1.6	72 (26–97)	-	I–II: 42% III–IV: 58%	Poor R-IPI 53%	R-CHOP or R-CHOP-like 83%	31 (N = 184 subcohort R-CHOP or R-CHOP-like)	8%	-	-	12%	-	-	-	-	-
Lim et al. [41]	South Korea	Retrospective	2009–2014	512	1.2	57 (47–67)	-	I–II: 56% III–IV: 44%	3–5: 32%	R-CHOP	52 (1–127)	12% (bilateral)	11%	1%	-	-	-	13%	8%	2%
																			2% heterogeneous	
Saiki et al. [42]	Japan	Retrospective	2008–2017	84	1.2	70 (19–86)	-	I–II: 47% III–IV: 53%	-	Mostly R-CHOP	-	26%	19%	7%	-	-	-	19%	8%	11%
Martin-Moro et al. [43]	Spain	Retrospective	2013–2017	82 (38 PET data)	1.2	63 (33–85)	GCB: 49% Non-GCB: 51%	I–II: 50% III–IV: 50%	aaIPI 2–3: 41%	R-CHOP or R-CHOP-like	33 (NR)	13%	-	-	24%	-	-	16%	-	-
Al-Sabbagh et al. [44]	Qatar	Retrospective	2013–2017	89	2.6	48 (18–77)	-	I–II: 39% III–IV: 61%	-	-	-	13%	-	-	-	-	-	26% (focal and heterogeneous)	-	-
Min et al. [45]	South Korea	Retrospective	2009–2016	600	1.4	59 (17–88)	GCB: 36% Non-GCB: 64%	I–II: 44% III–IV: 56%	3–5: 50%	R-CHOP	50 (0.2–123)	15% (bilateral)	-	-	-	-	-	16%	10%	3%
																			Both 3%	
Kandeel et al. [46]	Egypt	Retrospective	2015–2018	88	0.7	50 (21–70)	GCB: 43% Non-GCB: 57%	I–II: 14% III–IV: 86%	-	Mostly R-CHOP	11 (2–20)	25%	-	-	-	-	-	-	30%	-
Alonso-Alvarez et al. [47]	Spain	Retrospective	1999–2014	232	1	66% >60yo	GCB: 37% Non-GCB: 63%	I–II: 29% III–IV: 71%	Poor R-IPI: 41%	81% R-CHOP or R-CHOP-like	58 (1–152)	25%	16%	9%	25%	10%	15%	-	-	-
Wang et al. [48]	South Korea	Retrospective	2011–2017	140	1.4	65 (22–86)	GCB: 9% Non-GCB: 91%	I–II: 0% III–IV: 100%	>3: 45%	R-CHOP	49 (1–98)	36% (bilateral)	-	-	-	-	-	-	31%	-
Greenbaum et al. [49]	Israel	Retrospective	2005–2014	81	1.5	65 (23–87)	-	I–II: 38% III–IV: 62%	Median 3	91% R-CHOP	-	26%	-	-	63%	-	-	-	-	-
Chen et al. [16]	China	Retrospective	2007–2016	193	0.9	58 (14–87)	GCB: 32% Non-GCB: 68%	I–II: 44% III–IV: 56%	3–5: 43%	R-CHOP	30 (12–124)	7%	-	-	-	-	-	24%	15%	9%
Chen-Liang et al. [50]	Spain	Retrospective	2007–2015	268	1	61 (18–85)	-	I–II: 25% III–IV: 75%	3–5: 42%	76% R-CHOP	25 (1–91)	13%	-	-	-	-	-	22%	17%	6%
Vishnu et al. [51]	USA	Retrospective	2004–2013	99	1.7	62 (24–88)	-	-	Poor R-IPI: 24%	-	91 (28–140)	14%	-	-	-	-	-	24%	-	-
Alzahrani et al. [33]	Canada and Denmark	Retrospective	2007–2013	530	1.2	65 (16–90)	-	I–II: 37% III–IV: 63%	3–5: 43%	-	24 (3–78)	16%	10%	7%	-	-	-	-	28%	-
Liang et al. [52]	China	Retrospective	2005–2014	169	1.3	55 (18–85)	GCB: 40% Non-GCB: 60%	I–II: 36% III–IV: 64%	4–5: 17%	R-CHOP (60%) or DA-EPOCH-R (40%)	38 (12–113)	12% (some bilateral)	-	-	-	-	-	21%	20%	2%
Cerci et al. [20]	Brazil, Chile, Hungary, India, Italy, South Korea, Philippines, and Thailand	Prospective	2008–2011	327	1.1	55 (IQR 44–63)	-	I–II: 36% III–IV: 64%	3–5: 34%	R-CHOP recommended	35 (NR)	11%	-	-	-	-	-	26%	21%	6%
Adams et al. [53]	The Netherlands	Retrospective	2007–2013	78	1.2	69 (33–88)	-	I–II: 23% III–IV: 77%	NCCN-IPI >3: 71%	91% R-CHOP	28 (5–74)	21% (3% undetermined)	14%	4%	-	-	-	44%	39%	5%
Arima et al. [54]	Japan	Retrospective	2006–2011	96 ^‡^	1.5	69 (22–89)	-	I–II: 39% III–IV: 61%	3–5: 52%	R-CHOP	36 (1–72)	20%	15%	5%	28%	-	-	-	-	-
Berthet et al. [10]	France	Retrospective	2006–2011	133	1	57 (18–87)	-	I–II: 26% III–IV: 74%	3–5: 40%	R-CHOP or R-CHOP-like	24 (1–67)	6%	4%	2%	-	-	-	32%	24%	8%
Khan et al. [35]	United Kingdom	Retrospective	2005–2012	130	1.5	59 (22–87)	-	I–II: 45% III–IV: 55%	3–5: 40%	95% R-CHOP	-	11%	11%	0%	-	-	-	25%	21.5%	1.5%
																			Both 2%	
Hong et al. [36]	South Korea	Retrospective	2007–2011	89	0.8	59 (26–83)	GCB: 49% Non-GCB: 51%	I–II: 47% III–IV: 52%	Poor R-IPI: 35%	R-CHOP	16 (NR)	16% (bilateral)	10%	6%	-	-	-	19%	11%	8%
Cortes Romera et al. [55]	Spain	-	2004–2010	84	1	63 (19–78)	-	I–II: 50% III–IV: 50%	-	R-CHOP	NR (9–34)	-	19%	-	-	-	-	29%	23%	6%

aa: age-adjusted; COO: cell-of-origin; DLBCL: diffuse large B-cell lymphoma; GCB: germinal center B; IPI: International Prognostic Index; IQR: interquartile range; NCCN: National Comprehensive Cancer Network; PET-FDG: positron emission tomography-fluorodeoxyglucose; R-IPI: revised IPI; Ref.: reference; yo: years old. * References are ordered according to the sequence obtained from the bibliographic search. ^†^ Taking into account cases with available data to calculate the percentages. ^‡^ After excluding 21 cases with FCM not performed.

**Table 2 diagnostics-14-00658-t002:** Concordance analysis between BMB, FCM, and PET in the assessment of BMi at DLBCL diagnosis.

Ref.	Concordant Results						Discordant Results						Cohen’s Kappa Index		
	BMB+/FCM+	BMB+/PET+	FCM+/PET+	BMB−/FCM−	BMB−/PET−	FCM−/PET−	BMB+/FCM−	BMB+/PET−	BMB−/FCM+	BMB−/PET+	FCM+/PET−	FCM−/PET+	BMB/FCM	BMB/PET	FCM/PET
Bo et al. [37]	-	16% (23% after second punction)	14% (20% after second punction)	-	71%	72%	-	5%	-	10% (3% after second punction)	4%	12% (6% after second punction)	-	0.59 (0.8 after second punction)	0.54 (0.74 after second punction)
Han et al. [38]	-	7%	-	-	76%	-	-	6%	-	11%	-	-	-	0.36 *	-
Kim et al. [39]	-	5%	-	-	79%	-	-	4%	-	12%	-	-	-	0.32 *	-
Okamoto et al. [40]	3%	-	-	83%	-	-	5%	-	9%	-	-	-	0.25	-	-
Lim et al. [41]	-	7%	-	-	83%	-	-	5%	-	6%	-	-	-	0.51 *	-
Saiki et al. [42]	-	10%	-	-	64%	-	-	17%	-	10%	-	-	-	0.26 *	-
Martín-Moro et al. [43]	13%	5%	8%	76%	79%	71%	0%	5%	15%	11%	13%	8%	0.65 *	0.31 *	0.30 *
Al-Sabbagh et al. [44]	-	12%	-	-	73%	-	-	1%	-	13%	-	-	-	0.55 *	-
Min et al. [45]	-	9%	-	-	76%	-	-	7%	-	8%	-	-	-	0.43 *	-
Kandeel et al. [46]	-	14%	-	-	59%	-	-	11%	-	16%	-	-	-	0.31 *	-
Alonso-Alvarez et al. [47]	18%	-	-	69%	-	-	6%	-	7%	-	-	-	0.65 *	-	-
Wang et al. [48]	-	21%	-	-	54%	-	-	14% ^†^	-	10%	-	-	-	0.46 *	-
Greenbaum et al. [49]	26%	-	-	37%	-	-	0%	-	37%	-	-	-	0.4 *	-	-
Chen et al. [16]	-	7%	-	-	76%	-	-	1%	-	17%	-	-	-	0.36 *	-
Chen-Liang et al. [50]	-	9%	-	-	74%	-	-	4%	-	13%	-	-	-	0.41 *	-
Vishnu et al. [51]	-	12%	-	-	74%	-	-	2%	-	12%	-	-	-	0.55 *	-
Alzahrani et al. [33]	-	9%	-	-	66%	-	-	7%	-	18%	-	-	-	0.30 *	-
Liang et al. [52]	-	11%	-	-	78%	-	-	1%	-	10%	-	-	-	0.59 *	-
Cerci et al. [20]	-	8%	-	-	71%	-	-	3%	-	19%	-	-	-	0.31 *	-
Adams et al. [53]	-	14%	-	-	50%	-	-	6%	-	29%	-	-	-	0.22 *	-
Arima et al. [54]	18%	-	-	70%	-	-	2%	-	10%	-	-	-	0.66 *	-	-
Berthet et al. [10]	-	5% (4% subcohort PET focal)	-	-	67% (74% subcohort PET focal)	-	-	1% (2% subcohort PET focal)	-	27% (20% subcohort PET focal)	-	-	-	0.19 * (0.23 * subcohort PET focal)	-
Khan et al. [35]	-	9%	-	-	73%	-	-	2%	-	16%	-	-	-	0.42 *	-
Hong et al. [36]	-	8%	-	-	73%	-	-	8%	-	11%	-	-	-	0.34 *	-
Cortes Romera et al. [55]	-	18%	-	-	70%	-	-	1%	-	11%	-	-	-	0.68 *	-

BMB: bone marrow biopsy; FCM: flow cytometry; PET: positron emission tomography; Ref.: reference. * Not reported; calculated according to data extracted from each study. ^†^ All cases presented a diffuse BM pattern by PET, considered negative BMi by the authors.

**Table 3 diagnostics-14-00658-t003:** Diagnostic accuracy of BMB and PET for detecting BMi at DLBCL diagnosis.

Ref.	Definition of True Positive BMi	Sensitivity (CI 95%)			Specificity (CI 95%)			Positive/Negative Predictive Value			Youden’s Index/Diagnostic Accuracy		
		BMB	PET	PET Focal	BMB	PET	PET Focal	BMB	PET	PET Focal	BMB	PET	PET Focal
Bo et al. [37]	After first direct BM study (BMB and FCM)	-	62% (43–78)	-	-	93% (86–97)	-	-	-	-	-	0.55% (YI)	-
	After second direct BM study (BMB and FCM)	-	92% (76–98)	-	-		-	-	-	-	-	0.86% (YI)	-
Lim et al. [41]	Bilateral BMB	-	59% (NR)	-	-	94%	-	-	55%/95%	-	-	90% (DA)	-
Saiki et al. [42]	BMB	-	36% (NR)	-	-	87% (NR)	-	-	50%/79%	-	-	-	-
Al-Sabbagh et al. [44]	BMB or PET+ any of guided biopsy confirmation/MRI/focal uptake/FDG disappearance with treatment	50% (29–71)	96% (79–100)	-	100% (94–100)	100% (95–100)	-	100%/84%	100%/98%	-	87% (DA)	99% (DA)	-
Min et al. [45]	Bilateral BMB	-	52%	-	-	91%	-	-	-	-	-	-	-
Kandeel et al. [46]	BMB or PET+ any of concordant morphologic changes by CT/FDG disappearance with treatment/concordant FDG progression on follow-up	69% (NR)	-	67% (NR)	100% (NR)	-	90% (NR)	100%/85%	-	77%/84%	89% (DA)	-	82% (DA)
Chen et al. [16]	BMB or PET focal	44% (NR)	88% (NR)	-	-	-	-	NR/90%	NR/98%	-	91% (DA)	98% (DA)	-
Chen-Liang et al. [50]	BMB or PET	40% (27–53)	69% (52–85)	-	95% (91–98)	85% (80–89)	-	NR/85%	NR/95%	-	83% (DA)/0.4 (YI)	83% (DA)/0.5 (YI)	
Vishnu et al. [51]	BM aspirate and trephine biopsy	-	86% (56–97)	-	-	86% (76–92)	-	-	50%/98%	-	-	86% (DA)	
Alzahrani et al. [33]	BMB or PET	48% (41–56)	-	81% (74–86)	-	-	-	NR/79%	-	NR/91%	-	-	-
	BMB	-	-	60% (49–70)	-	-	79% (75–83)	-	-	36%/91%	-	-	-
	BMB concordant	-	-	77% (63–87)	-	-	79% (75–83)	-	-	29%/97%	-	-	-
	PET focal	36% (28–44)	-	-	91% (88–94)	-	-	60%/79%	-	-	-	-	-
Berthet et al. [10]	BMB or PET focal + confirmed by guided biopsy/targeted MRI/after chemotherapy by concomitant disappearance of uptake	24% (9–39)	-	94% (86–100)	100% (100–100)	-	99% (97–100)	100%/80%	-	97%/98%	81% (DA)	-	98% (DA)
Khan et al. [35]	BMB or PET	40% (NR)	94% (NR)	-	100% (NR)	100% (NR)	-	NR	NR	-	84% (DA)	99% (DA)	-
Cortes Romera et al. [55]	BMB	-	95%	-	-	86%	-	-	54%/99%	-	-	87% (DA)	

BM: bone marrow; BMB: bone marrow biopsy; CI: confidence interval; DA: diagnostic accuracy; FCM: flow cytometry; MRI: magnetic resonance imaging; NR: not reported; PET: positron emission tomography; Ref.: reference; YI: Youden’s index.

**Table 4 diagnostics-14-00658-t004:** Prognostic impact of BMi at DLBCL diagnosis according to BMB, FCM, and PET in comparative studies.

Ref.	N	Adverse Factor Related to BM Assessment (Compared with Its Complementary Good Prognosis Reference in Each Case)		Survival Endpoints					
				Event-Free Survival or Progression-Free Survival			Overall Survival		
				Prognosis	UV HR (CI 95%)	MV HR (CI 95%)	Prognosis	UV HR (CI 95%)	MV HR (CI 95%)
Okamoto et al. * [40]	184 (subcohort R-CHOP or R-CHOP-like)	BMB+		20% (2-year) C-index 0.68	4.3 (2.1–8.8)	2.3 (1.3–6.5) ^1^	24% (2-year) C-index 0.74	5.2 (2.6–10.5)	3 (1.3–6.8) ^1^
		FCM+		C-index 0.70	3.1 (1.6–5.8)	2.8 (1.4–5.2) ^1^	C-index 0.74	2.2 (1.1–4.5)	1.9 (0.9–3.7) ^1^
Lim et al. ^‡^ [41]	512	BMB+		-	3.1 (1.8–5.4)	1.7 (1.1–2.6) ^2^	52% (2-year) 37% (5-year)	-	-
		PET	PET+	29% (2-year)	2.8 (1.6–4.7)	1.7 (1.1–2.6) ^2^	-	-	-
			PET+ focal	-	2.1 (1.1–3.9)	-	-	-	-
			PET+ diffuse	-	3.1 (0.8–11)	-	-	-	-
			PET+ heterogeneous	-	4.2 (1–16.9)	-	-	-	-
Martin-Moro et al. * [43]	82	BMB+/FCM+		27% (18-month)	2.2 (1.4–3.3)	-	55% (18-month)	1.9 (1.2–3)	-
		BMB−/FCM+		23% (18-month)	4.9 (1.7–14.2)	-	46% (18-month)	4.4 (1.5–12.4)	-
		FCM+		-	4.8 (2.3–10)	1.9 (1.3–2.9) ^3^ 2 (1.3–3.1) ^4^	-	3.8 (1.8–8.3)	1.7 (1.1–2.7) ^3^ 1.7 (1.1–2.7) ^4^
Min et al. * [45]	600	BMB+		-	-	-	56% (4-year)	-	1 (0.8–1.3) ^5^
		PET+		43% (4-year)	-	-	65% (4-year)	-	-
Kandeel et al. ^•^ [46]	88	BMB+		64% (18-month)	1.4 (0.5–3.5)	-	-	-	-
		PET+ focal		73% (18-month)	1 (0.4–2.5)	-	-	-	-
Alonso-Alvarez et al. * [47]	189 (subcohort R-CHOP or R-CHOP-like)	Concordant	BMB+ concordant	32% (5-year)	-	2.2 (1.1–4.3) ^6^	51% (5-year)	-	1.6 (0.7–3.4) ^6^
			BMB+ concordant GCB	25% (5-year)	-	2.9 (1–8.7) ^7^	38% (5-year)	-	1.2 (0.3–4.4) ^7^
			BMB+ concordant non-GCB	33% (5-year)	-	3 (1.4–6.4) ^7^	49% (5-year)	-	1.6 (0.7–3.9) ^7^
		Discordant	BMB/FCM+ discordant	62% (5-year)	-	1.5 (0.7–3) ^6^	73% (5-year)	-	1.5 (0.7–3.2) ^6^
			BMB/FCM+ discordant GCB	76% (5-year)	-	0.7 (0.1–3) ^7^	76% (5-year)	-	0.7 (0.1–3.4) ^7^
			BMB/FCM+ discordant non-GCB	46% (5-year)	-	1.9 (0.9–4.2) ^7^	63% (5-year)	-	1.6 (0.6–3.7) ^7^
Wang et al. ^†^ [48]	140 (all cases advanced stage)	BMB+		-	2.3 (1.4–3.9)	2.3 (1.4–3.9) ^8^	-	1.7 (1–2.8)	Not significant ^8^
		PET+ focal		-	2.1 (1.3–3.6)	Not significant ^8^	-	1.8 (1.1–3)	1.9 (1.1–3.1) ^8^
		BMB+/PET+		7 months (median)	-	-	12 months (median)	-	-
		BMB+/PET-		14 months (median)	-	-	27 months (median)	-	-
		BMB−/PET+		26 months (median)	-	-	31 months (median)	-	-
Greenbaum et al. ^†^ [49]	81	BMB+/FCM+		67 months (median) 47% (1-year) 36% (5-year) 24% (7-year)	-	-	54 months (median) 43% (1-year) 32% (5-year) 32% (7-year)	-	-
		BMB−/FCM+		77 months (median) 61% (1-year) 56% (5-year) 49% (7-year)	-	-	77 months (median) 60% (1-year) 55% (5-year) 54% (7-year)	-	-
		FCM+		-	-	2.6 (1–6.8) ^9^	-	-	1.4 (0.7–3) ^9^
Chen et al. ^†^ [16]	193	BMB+		61% (3-year)	1.8 (0.7–4.6)	-	61% (3-year)	4 (1.3–11.9)	-
		PET+ focal		33% (3-year)	4.4 (2.4–8.2)	2.3 (1.1–4.7) ^10^	69% (3-year)	3.7 (1.5–9.3)	Not significant ^10^
		PET+ diffuse		81% (3-year)	0.9 (0.3–2.4)	-	94% (3-year)	0.6 (0.1–4.3)	-
		BMB−/PET+ focal		0% (3-year)	-	-	77% (3-year)	-	-
		BMB−/PET+ diffuse		90% (3-year)	-	-	100% (3-year)	-	-
Chen-Liang et al. * [50]	203 (subcohort R-CHOP)	BMB+		-	*p* < 0.001	3.6 (1.7–7.6) ^11^	-	*p* = 0.326	
		PET+		-	*p* = 0.121	*p* > 0.15 ^11^	-	*p* = 0.018	*p* > 0.15 ^11^
Vishnu et al. [51]	99 (not reported treatment approach)	BMB+		-	-	-	65 months (median) 80% (2-year) 66% (5-year)	-	-
		PET+		-	-	-	67 months (median) 83% (2-year) 79% (5-year)	-	-
Alzahrani et al. * [33]	256 (subcohort Ann Arbor IV)	BMB+ or PET+		57% (2-year)	-	-	65% (2-year)	-	-
		PET+		53% (2-year)	-	-	63% (2-year)	-	-
Liang et al. ^†^ [52]	169	BMB+		-	4.5 (2.5–8)	NS ^12^	-	6.2 (3.1–12.7)	NS ^12^
		PET+		-	4 (2.3–6.6)	NS ^12^	-	6.7 (3.4–13.3)	2.9 (1.2–7) ^12^
	68 (subcohort Ann Arbor IV)	PET+		29% (3-year)	-	-	44% (3-year)	-	-
Cerci et al. ^•^ [20]	327	BMB+		56% (2-year)	2.2 (1.3–3.3)	-	68% (2-year)	-	-
		BMB+/PET+		45% (2-year)	2.7 (1.5–4.8)	1.6 (0.8–3.1) ^13^	55% (2-year)	3.9 (1.9–8.1)	2.3 (1–5) ^13^
		BMB+/PET+ focal		46% (2-year)	2.5 (1.1–5.5)	-	57% (2-year)	3 (1.1–8.4)	-
		BMB+/PET-		80% (2-year)	-	1.1 (0.4–3.2) ^13^	100% (2-year)	-	0.5 (0.1–3.7) ^13^
		BMB−/PET+		81% (2-year)	-	0.7 (0.4–1.3) ^13^	88% (2-year)	-	0.7 (0.3–1.7) ^13^
		BMB−/PET+ focal		78% (2-year)	-	-	87% (2-year)	-	
Adams et al. * [53]	71 (subcohort R-CHOP)	BMB+		-	3.2 (1.3–7.6)	3.3 (1.3–8.6) ^14^	-	3.5 (1.3–9.3)	4.5 (1.6–12.4) ^14^
		PET+		-	0.9 (0.4–2.1)	-	-	0.7 (0.3–2.2)	-
Berthet et al. * [10]	133	BMB+		38% (2-year)	4.9 (1.6–14.6)	2.2 (0.8–6) ^15^	63% (2-year)	4.1 (1.4–12.4)	2.7 (0.9–8.2) ^15^
		PET+		63% (2-year)	2.9 (1.2–7)	2.5 (1.2–5.3) ^15^	76% (2-year)	2.8 (1.2–6.8)	2.2 (0.9–5.3) ^15^
Khan et al. ^†^ [35]	44 (subcohort Ann Arbor IV)	BMB+		-	3.7 (1.6–8.8)	-	-	3.9 (1.5–10)	-
		PET+		-	0.8 (0.3–2)	-	-	0.9 (0.3–2.5)	-
		PET+/BMB−		-	0.4 (0.1–1.2)	-	-	0.4 (0.1–1.6)	-
Hong et al. * [36]	89	BMB+		37% (2-year)	-	-	36% (2-year)	7.9 (3.2–19.6)	8.7 (3.2–23.5) ^16^
		PET+		63% (2-year)	-	-	59% (2-year)	2 (0.8–5.3)	-
		BMB+/PET+		38% (2-year)	-	-	38% (2-year)	-	-
		BMB+/PET−		36% (2-year)	-	-	36% (2-year)	-	-
		BMB−/PET+		79% (2-year)	-	-	73% (2-year)	-	-

BMB: bone marrow biopsy; BMi: bone marrow involvement; CI: confidence interval; COO: cell-of-origin; ECOG: Eastern Cooperative Oncology Group; FCM: flow cytometry; FDG: fluorodeoxyglucose; HR: hazard ratio; LDH: lactate dehydrogenase; NCCN: National Comprehensive Cancer Network; IPI: International Prognostic Index; MV: multivariate; PET: positron emission tomography; R-IPI: revised IPI; Ref.: reference; UV: univariate. Survival endpoint: * Event-free survival (EFS): time from diagnosis to first disease progression, relapse, or death by any cause. ^†^ Progression-free survival (PFS): time from diagnosis to first disease progression or relapse. ^‡^ Lymphoma-specific survival (LSS): time from diagnosis to death caused by relapse/refractory disease or treatment-related complications of lymphoma. ^•^ The definition of EFS/PFS is not reported. Overall survival was defined as the time from diagnosis to death by any cause in all studies. Adverse variables reported as included in each multivariate analysis include the following: ^1^ Adjusting for IPI factors: age, LDH, PS > 1, stage > II, and extranodal site. ^2^ Includes age > 60, IPI 4–5, bulky, biopsy-confirmed BMI, and any pattern of FDG+ BM. ^3^ Includes FCM BMi, COO Hans, and IPI. ^4^ Includes FCM BMi and IPI. ^5^ Includes age ≥ 60 years, Ann Arbor III–IV or II bulky, high LDH, ≥2 extranodal sites, ECOG 2–4, BMi by BMB, COO Hans, and complex karyotype. ^6^ Includes R-IPI and concordant/discordant BMi. ^7^ Includes R-IPI and concordant/discordant BMi with COO. ^8^ Includes male sex, age, BMi by BMB, BMi by PET focal, Ann Arbor IV (vs. III), IPI > 3, and B symptoms. ^9^ Includes IPI, BMi by FCM, sex male, and hemoglobin. ^10^ Includes advanced stage and PET+ focal. ^11^ Includes male sex, age > 60, ECOG > 1, elevated beta-2 microglobulin, elevated LDH, BMi by PET, and BMi by BMB. ^12^ Includes BMi by BMB, BMi by PET, IPI >2, Ann Arbor III–IV, ECOG > 1, elevated LDH, and extranodal > 1. ^13^ Includes BMi by a combination of BMB and PET (BMB+/PET+, BMB+/PET−, or BMB−/PET+, respectively), age > 60 years, Ann Arbor III–IV, high LDH, ≥2 extranodal sites, ECOG 2–4, and omission of rituximab therapy. ^14^ Includes BMi by BMB, BMi by PET, metabolic parameters by PET, and high-risk NCCN-IPI. ^15^ Includes BMi by BMB and PET, age > 60 years, Ann Arbor III–IV, high LDH, ≥2 extranodal sites, ECOG 2–4, and IPI > 2. ^16.^ Includes BMi by BMB, elevated LDH, Ann Arbor III–IV, ECOG > 1, and extranodal > 1.

## Data Availability

All the studies used in this study are published in the literature.

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
