# Peer review of "Comparative Review of the Current and Future Strategies to Evaluate Bone Marrow Infiltration at Diffuse Large B-Cell Lymphoma Diagnosis"

_diagnostics, 2024, doi:10.3390/diagnostics14060658_

Round 1

Reviewer 1 Report

Comments and Suggestions for Authors

The draft submitted by Martin-Moro et al. made a systemic review of the prognostic values of current medical interventions in evaluating BMi for DLBCL patients. For the past few decades since rituximab entered the market, DLBCL diagnosis and characterization become essential for the options of treatments. Despite using BMi for DLBCL diagnosis being controversial, bone marrow biopsy (BMB) is considered the gold standard to aid the diagnosis for a certain period of time. With the introduction of FDG PET imaging, more diagnoses rely on the metabolic activity of cancer cells, leading to a more comprehensive bodywide screening. Therefore, this study intends to compare the diagnostic/prognostic value of BMB, FDG imaging, and flowcytometry (FCM) analysis for DLBCL cases, in a systemic review way. Althought the draft is detailed drafted, a few suggestions are made:

1) It is still unclear which diagnostic method is more feasible, or at least, has a better diagnostic accuracy. According to the provided tables, the accuracy is diverse, and there is no clear cut-off value for valid diagnosis discussed. It is unclear how and what measurements should be put into consideration in a selection of methods. The authors may provide more specific objectives in the introductions. 

2) The tables are comprehensively listed and annotated. However, the results are simply reporting rather than allowing a brief summary in each segment. Considering the size of discussion, it might be helpful if the authors can highlight the results by each paragraph to enhance the reading experience. 

3) It's understandable but interesting that the conclusion is still puzzling in method selection. It is expected to have some solid suggestions or specific findings from the systemic review. 

Comments on the Quality of English Language

Some typos are noticed in the figure, and plenty of grammatical errors are noticed in both subtitles and texts. Substantial language editing is suggested. 

Author Response

  1. Response to Reviewer 1 comments:
  • Comments and Suggestions for Authors:
    • It is still unclear which diagnostic method is more feasible, or at least, has a better diagnostic accuracy. According to the provided tables, the accuracy is diverse, and there is no clear cut-off value for valid diagnosis discussed. It is unclear how and what measurements should be put into consideration in a selection of methods. The authors may provide more specific objectives in the introductions. -> As suggested by the reviewer, we modified the Introduction to clarify the purposes of the review. The final paragraph related to this comment is: “On many occasions, discrepancies between the different techniques occur when assessing BMi in DLBCL, and the diagnostic accuracy and definition of a true positive BMi is still controversial. Furthermore, the surveillance implication of BMi is still not clear and some studies have combined and compared the results of different techniques in an attempt to solve this topic. Our aim is to perform a comparative review of the three techniques most widely performed when evaluating BMi at DLBCL diagnosis: (BMB, PET, and FCM). We analysed the rate of BMi by each technique and their diagnostic accuracy, described the concordance and discordance among procedures in detecting BMi, and evaluated the prognostic implication of the BMi by each technique when compared or combined with the others. When appropriate and available, data about each type of result was also collected, thus, concordant or discordant BMi by BMB and FCM, and focal or diffuse pattern of BMi by PET-FDG”.
    • The tables are comprehensively listed and annotated. However, the results are simply reporting rather than allowing a brief summary in each segment. Considering the size of discussion, it might be helpful if the authors can highlight the results by each paragraph to enhance the reading experience. -> We agree with the reviewer and strongly appreciate this comment. Therefore, we added a brief summary of each section in Results to facilitate reading:
      • 1. Summary of the section: all reports including at least two techniques for the evaluation of BMi at DLBCL diagnosis reported information about BMB. FCM and PET showed higher rates of BMi (24.5% and 22%, respectively) than BMB (15%).
      • 2. Summary of the section: a substantial concordance has been described between BMB and FCM in evaluating BMi at DLBCL diagnosis, while concordance between BMB and PET was fair. Discordance among techniques was common.
      • 3. Summary of the section: PET-FDG showed higher sensitivity and BMB higher specificity when assessing BMi at DLBCL diagnosis, although high heterogeneity was seen among reports.
      • 4. Summary of the section: BMi detected by FCM is associated with worse outcomes in newly DLBCL, even independently from BMB findings. The prognostic impact of BMi by PET is less clear.
    • It's understandable but interesting that the conclusion is still puzzling in method selection. It is expected to have some solid suggestions or specific findings from the systemic review. -> This manuscript cannot be considered as a systematic review as the proposal is describing the current situation and perspectives about the field, therefore, the aim is not answering a focused clinical question. An example of systematic review regarding this field was published by Adams HJ et al in 2014 (doi: 10.1007/s00259-013-2623-4), analyzing the diagnostic accuracy of FDG PET/CT for the detection of BMi in DLBCL. As our study includes different topics (concordance, diagnostic accuracy, and prognostic impact) and techniques (BMB, FCM and PET) about BMi in DLBCL it is not considered a systematic review but a literature review, and more solid conclusions cannot be made. Furthermore, the development process carried out for our manuscript is different from that when performing a systematic review. Despite all this, we tried to make specific suggestions in Conclusions, such as “Both the BMB, the FCM, and the PET-FDG are complementary tools that provide different and clinically relevant information” and “There is still not enough evidence to recommend avoiding the direct BM assessment at baseline evaluation of DLBCL”.
  • Quality of English Language: “Moderate editing of English language required”. Some typos are noticed in the figure, and plenty of grammatical errors are noticed in both subtitles and texts. Substantial language editing is suggested. à A substantial editing was done to solve language mistakes, with a special focus on figures, titles, and subtitles. The text and Figure 1 were modified accordingly.

Reviewer 2 Report

Comments and Suggestions for Authors

This is a very interesting review article comparing different manuscripts with different techniques in diagnosing BMi in DLBCL. The author mainly focused on comparing the results among BMB, FCM and PET. Finally, the author comes up a conclusion that all three tools provided slightly different  results focus on different clinical outcomes. 

Here are some suggestions for authors to consider for this manuscript.

1. please make sure it is written as a review or article (based on literature research purpose). This paper looks very similar to a research article rather than a review, and it presents the results (numbers and percents of article number and results) of the comparison of the manuscript. 

2. When describing the results of paper comparison, are the results from different techniques correlated? or are they not correlated for all the results s they have measured while they total focused on different clincal insights. 

Author Response

  1. Response to Reviewer 2 comments:
  • Comments and Suggestions for Authors:
    • Please make sure it is written as a review or article (based on literature research purpose). This paper looks very similar to a research article rather than a review, and it presents the results (numbers and percents of article number and results) of the comparison of the manuscript. -> We thank the reviewer for pointing this out. This text cannot be considered as an article as it is not an original research manuscript, and we believe that the manuscript should be considered as a review as it offers a comprehensive analysis of the existing literature within the field of study.
    • When describing the results of paper comparison, are the results from different techniques correlated? or are they not correlated for all the results s they have measured while they total focused on different clincal insights. -> The manuscript compares the BM assessment by BMB, FCM and PET, including research articles reporting at least two techniques. Only data considered as clinically relevant according to this review’s aims were included in the research articles. As presented in the flow chart (Figure 1), manuscripts reporting data not focused on BMi in DLBCL were studied. Correlations among techniques were not reported in all papers, but we made an effort to extract as much information as possible to describe those correlations. As a result, in Table 1 we described the rate of BMi by each technique with a comparison between them, and in Table 2 we performed a direct correlation among techniques presenting the reported concordance and discordance among BM assessment in DLBCL. The purpose of Tables 3 and 4 was to correlate indirectly results about diagnostic accuracy and prognostic significance.
  • Quality of English Language: “English language fine. No issues detected”. à No review required.
